# Chaotic Search-and-Rescue-Optimization-Based Multi-Hop Data Transmission Protocol for Underwater Wireless Sensor Networks

**DOI:** 10.3390/s22082867

**Published:** 2022-04-08

**Authors:** Durairaj Anuradha, Neelakandan Subramani, Osamah Ibrahim Khalaf, Youseef Alotaibi, Saleh Alghamdi, Manjula Rajagopal

**Affiliations:** 1Department of Computer Science and Business Systems, Panimalar Engineering College, Chennai 600123, India; 2Department of Computer Science and Engineering, R.M.K Engineering College, Chennai 600123, India; snk.cse@rmkec.ac.in; 3Al-Nahrain Nano Renewable Energy Research Center, Al-Nahrain University, Baghdad 64074, Iraq; usama.ibrahem@coie-nahrain.edu.iq; 4Department of Computer Science, College of Computer and Information Systems, Umm Al-Qura University, Makkah 21955, Saudi Arabia; yaotaibi@uqu.edu.sa; 5Department of Information Technology, College of Computers and Information Technology, Taif University, Taif 21944, Saudi Arabia; s.algamedi@tu.edu.sa; 6Department of Computer Science and Engineering, School of Computing, SRM Institute of Science and Technology, Kattankulathur, Chennai 603203, India; manjular1@srmist.edu.in

**Keywords:** data transmission, routing, search and rescue optimization, chaotic concept, fitness function, underwater wireless sensor network

## Abstract

Underwater wireless sensor networks (UWSNs) have applications in several fields, such as disaster management, underwater navigation, and environment monitoring. Since the nodes in UWSNs are restricted to inbuilt batteries, the effective utilization of available energy becomes essential. Clustering and routing approaches can be employed as energy-efficient solutions for UWSNs. However, the cluster-based routing techniques developed for conventional wireless networks cannot be employed for a UWSN because of the low bandwidth, spread stay, underwater current, and error probability. To resolve these issues, this article introduces a novel chaotic search-and-rescue-optimization-based multi-hop data transmission (CSRO-MHDT) protocol for UWSNs. When using the CSRO-MHDT technique, cluster headers (*CH*s) are selected and clusters are prearranged, rendering a range of features, including remaining energy, intracluster distance, and intercluster detachment. Additionally, the chaotic search and rescue optimization (CSRO) algorithm is discussed, which is created by incorporating chaotic notions into the classic search and rescue optimization (SRO) algorithm. In addition, the CSRO-MHDT approach calculates a fitness function that takes residual energy, distance, and node degree into account, among other factors. A distinctive aspect of the paper is demonstrated by the development of the CSRO algorithm for route optimization, which was developed in-house. To validate the success of the CSRO-MHDT method, a sequence of tests were carried out, and the results showed the CSRO-MHDT method to have a packet delivery ratio (PDR) of 88%, whereas the energy-efficient clustering routing protocol (EECRP), the fuzzy C-means and moth–flame optimization (FCMMFO), the fuzzy scheme and particle swarm optimization (FBCPSO), the energy-efficient grid routing based on 3D cubes (EGRC), and the low-energy adaptive clustering hierarchy based on expected residual energy (LEACH-ERE) methods have reached lesser PDRs of 83%, 81%, 78%, 77%, and 75%, respectively, for 1000 rounds. The CSRO-MHDT technique resulted in higher values of number of packets received (NPR) under all rounds. For instance, with 50 rounds, the CSRO-MHDT technique attained a higher NPR of 3792%.

## 1. Introduction

An underwater wireless sensor network (UWSN) consists of several underwater wireless sensors distributed within the marine environment that assist wide-ranging applications, including navigation, surveillance, disaster prevention, data acquisition, and resource exploration [1,2,3]. All the sensors of a UWSN are armed with an acoustic modem since they employ acoustic signals to interact with one another [4,5,6]. This node is able to form a network without any architecture. The sensors are responsible for monitoring the underwater environments, namely the temperature, and transmit the gathered information to a sink node (SN) via a single hop or many hops [7,8,9]. The SN, placed on the aquatic surface, is capable of receiving the information from underwater sensors via acoustic signal and transmitting the collected information to terrestrial network devices via radio signal [10,11,12]. In underwater environments, the radio signal attenuates quickly and faces the absorption problem. Therefore, they are unsuitable for longer-distance underwater communication.

The complete upsurge is adapted at the time of submerged communication since it is lesser affected by scattering, absorption loss, and attenuation [13,14]. The acoustic signal propagates slowly causing higher propagation delay. Furthermore, there is another drawback of the underwater acoustic network, namely the high error rate, and the low bandwidth. Hence, it uses lot of energy to effectively transfer information packs in a UWSN and keeps the better presentation of a UWSN. Moreover, the sensors have constrained energy, and it is not easier to redistribute or recharge them [15,16,17]. Thus, the network lifetime and energy consumption are primary concerns in a UWSN.

The routing protocol for a UWSN deals with the path selection to transmit data packets to the surface destination in an effective and efficient manner [18,19]. In recent times, engineers, researchers, and scientists have employed the routing protocol to examine the underwater medium for various applications. The proposal of the routing protocol for UWSNs is of great significance. This protocol identifies a path from the bottom to the top of the water surface for ensuring system efficiency, based on the desired parameter [20,21]. Especially, the problems related to the underwater medium and at the time of packet forwarding are taken into account by this protocol for achieving the optimum efficiency of the network based on the intended purpose. For example, this protocol deals with the shadow zones, high propagation delay, constrained battery power, movement of the sensors with water current, reliable transmission of data packet at the time of unfavorable channel condition, severe noise, and interference [22]. 

The transmission of information through the multi-hop method was proved in [23] to be efficient in energy preservation in longer-distance transmission than the single-hop method. Therefore, it was necessary to discover the optimum multi-hop paths for achieving improved transmission efficacy, reduced packet loss ratio (PLR), and minimized energy consumption at the time of data transmission [24,25].

This article develops an effective chaotic search-and-rescue-optimization-based multi-hop data transmission (CSRO-MHDT) protocol for UWSNs to optimize energy efficiency and lifetime. At the initial stage, the CSRO-MHDT protocol involves the design of a weighted clustering approach (WCA) for the effectual selection of cluster heads (*CH*s) and cluster construction. In addition, the CSRO algorithm is presented by integrating the chaotic concepts into the traditional search and rescue optimization (SRO) algorithm. Moreover, the CSRO-MHDT technique has a suitability purpose, connecting remaining energy, reserves, and node degrees. For assessing the better outcomes of the CSRO-MHDT technique, a wide-ranging experimental analysis was carried out and the results were assessed under several aspects.

The following is the outline for the remainder of the paper. Refer to Section 2 for additional information on relevant works. Then, Section 3 delves into the specifics of the suggested paradigm. Then, Section 4 examines the performance of the suggested model, and Section 5 concludes the research.

## 2. Related Works

According to Durrani and colleagues [26], an adoptive, clustered-node routing method for a smart ocean underwater sensor network has been presented (SOSNET). The described approach makes use of a moth–flame optimizer (MFO) to calculate the near-optimal number of clusters that are required for routing to be effective. Moth migration toward the light is taken into consideration while using the MFO, a biologically inspired optimization technique. 

NR et al. [27] introduced the lion optimized cognitive acoustic network (LOCAN) for reducing packet delay and packet loss at the time of transmitting packets in a UWASN; these are caused by water column variation, including the Doppler effect and geometric spreading (GS). Doppler effects form because of sensor node movement and sea surface variations, including temperature and salinity. 

The authors in [28] developed a protocol-inspired method called the beckoning penguin swarm optimization protocol (BPSOP), which was inspired by the natural features of penguins. The foraging features of penguins are employed for finding the optimal route in a UWSN. 

The authors in [29] designed an energy-effectual and void-region-avoidance routing method. The idea of the GWO approach is utilized for selecting an optimal forwarder node. The presented method expands the lifetime of the network by balancing the network energy and preventing the void region. 

In [30], a new approach, called improved energy-balanced routing (IEBR), was developed for UWSNs. The presented method consists of two phases: data broadcast and routing establishment. To begin, a precise method for determining the transmission distance is developed in order to locate the neighbor at the optimal distance and to determine the submerged net connections [31,32,33]. Additionally, IEBR selects relays based on neighbor depth, minimizes hops in a connection-based depth threshold, and eliminates the data communication loop issue [34,35].

Rajeswari et al. [36] developed a cooperative ray optimization approach (CoROA), which assists in minimizing the packet loss and delay, which rise because of the geometric spreading and Doppler environments in underwater acoustic networks. The presented method is recognized for its effective performance in distinct environments, including spatial and temporal variations, where the throughput, battery life, and network lifetime are improved.

To improve the overall efficiency of a system, Fei et al. [37] suggested a hybrid clustering technique based on FCM and MFO with the goal of boosting system efficiency (FCMMFO) [38,39]. In order to achieve this, the researchers first used FCM to generate energy-efficient clusters, and then applied an optimization approach known as MFO to select the ideal *CH* for each cluster [40,41]. 

Yuhan Su et al. investigated the impacts of transmission-obliging broadcast and radio channel circumstances on submerged acoustic communication, and we calculated the best transmit power setting factor. No prior knowledge of the IoUT model was required to implement the proposed technique. The suggested technique can increase IoUT transmission performance according to the findings of the simulation. Compared with Q-learning, the DQN-based approach upsurges the joint information by 3% and decreases the outage likelihood by 40%. We plan to examine cooperative communications in IoUT in greater detail in the future. To increase cooperative communication in relay-assisted IoUT systems, novel reinforcement learning algorithms with low computational and communication overheads need to be developed.

Zhigang Jin et al. proposed a mobbing-avoidance routing protocol for UASNs based on strengthening knowledge. Through exploration, the RCAR protocol finds the best route to minimize congestion and save energy. The RCAR protocol extends reinforcement learning’s reward function with congestion and energy. To speed up the meeting process and to ensure the best routing decision, we used a simulated steering tube with changeable radius based on neighboring average residual energy.

Yougan Chen et al. offered the PB-ACR protocol for multi-hop UASNs, which includes node payload balancing as well as DCC. The energy consumption of each system node can be lowered by grouping packets based on the importance of the data included in them. While maintaining an appropriate balance between node payload and cooperative gain, the proposed PB-ACR protocol has the potential to improve system life and throughput when compared with the existing ACAR protocol.

In a study by Jianying Zhu et al., the proposed algorithm’s advantages diminish with the node count. This technique is also well-matched with the noncooperative ACOA-AFSA fusion algorithm, and it has a reasonable level of difficulty in time-variant marine surroundings. Therefore, the suggested ACOA-AFSA fusion DCC technique is more ideal for medium-sized networks, where it can boost data transmission reliability, while extending the life of a system. In future study, we will simplify the suggested algorithm’s hardware implementation to make it appropriate for other underwater acoustic networking applications. Table 1 shows the current state of wireless sensor networks (WSNs) using clustering and multi-hop routing.

## 3. The Proposed Model

In this study, a novel CSRO-MHDT method was developed for the optimal choice of routes for data transmission in a UWSN. Clustering is a well-known energy-saving strategy in sensor networks. The least-cost-clustering steering procedure (MCCP) is one of the clustering-based routing protocols (UWSNs). This parameter is composed of three important values: the total energy consumed by member nodes during data transfer to the *CH*, the total energy residual on the bunch head node and its associate bulges, and the distance between the cluster head node and the originating base station. The major intention of the CSRO-MHDT technique is to the reduce energy dissipation and enhance the lifetime of a UWSN. Primarily, the CSRO-MHDT technique involves WCA for the effectual choice of *CH*s. In addition, the CSRO-MHDT technique derives a fitness function and can effectually select the set of routes in a UWSN. Figure 1 presents the general procedure of the CSRO-MHDT method.

### 3.1. System Model

The system contains N dynamic bulges, which are sparse and arbitrarily dispersed during an L×L×L process. The data source is water-medium-sensed data. The data were gathered utilizing an underwater sensor. The current flow, pressure, and temperature are the identified parameters. The underwater sensor was prepared with an acoustic modem, which enables them to communicate with other nodes in the aquatic environment [42,43,44,45]. An SN is equipped with both a radio frequency (RF) and an acoustic modem for communication with the base station (BS) on the surface landmass; the SN’s acoustic modem receives data from underwater sensors, while the RF modem communicates with the base station (BS) by transmitting data through the base station. Despite the fact that it has a short transmission distance, the BS is capable of traversing a sensing field and collecting data from sensor nodes on the field. Each sensor node’s power consumption is lowered, since fewer relays are required to transmit the sensor’s message to the BS. It can be considered that the network condition is related to the networks. The topology varies rapidly due to the fact that underwater sensors are transferable, depending on water current velocities of around 1–3 m/s [46,47,48,49,50]. The network condition can be considered as follows:The bulges identify their place and the place of the SN in a primary situation.The node can develop the CH and the CM/relay.The *CH* rotates amongst the sensors to conserve energy.

Acoustic waves in an underwater broadcast medium have distinct characteristics from radio waves; hence, a WSN cannot be employed for underwater broadcasting networks. For the present study, we used the power consumption strategy of an underwater acoustic channel [51,52,53,54,55,56]. The energy required to transport *k* bits of information across a reserve, *d*, at an information amount, *R*, is computed as follows:(1)$$E_{Tx}\left(k,d\right)=k\times E_{elec}+\frac{k}{R}P_{tx}$$
where $E_{elec}$ implies the power utilization for routing 1 bit of information and $P_{Tx}$ stands for the transmitted power.

In order to received k bits of information, the receiver radio power utilization is provided under the following:(2)$$E_{Rx}\left(k\right)=kP_{r}$$

Assume that $P_{r}$ refers the constants dependent upon the devices. In order to fuse k bits of data, the power utilization is formulated as:(3)$$E_{DA}\left(k\right)=k\times E_{DA0}$$
where $E_{DA0}$ refers the energy used for fusing 1 bit of data, for instance, in use, as 5 nJ/bit. Fusing data is a frequent and successful method for removing data redundancy, shrinking data size, and lowering energy consumption. The data fusion is implemented in this research using an upgraded back propagation neural network (BPNN). Sensor nodes in UWSNs may collect information with great dismissal. When superfluous information is directed to the SN, wasteful energy ingesting occurs, resulting in the node’s premature death and the network’s lifespan being shortened. In comparison, if the CHNs integrate the data and send it to the SN, then significant energy savings may be realized [57,58]. While the node is mobile, caused by the water current, it can be located according to the random motion of the node under the functioning time. The current velocity is 1–3 $\frac{m}{s}$.

### 3.2. Process Involved in a WCA

The weight clustering technique defines the *CH* and utilized cluster infrastructure utilizing three measures: node degree ($ND_{i})$, ($RES_{i})$, and distance ($DIS_{i})$. For all the nodes, the weight, $P_{i}$, was computed as:(4)$$P_{i}=w_{1}*RES_{i}+w_{2}*DIS_{i}+w_{3}*ND_{i}$$

However, $w_{1}$, $w_{2}$, and $w_{3}$ signify the coefficient of model state, as follows: (5)$$w_{1}+w_{2}+w_{3}=1$$

The SN(x) for transmitting k bits of data to the receiver at distance, d, is calculated as follows:(6)$$RES=E-\left(E_{T}\left(k,d\right)+E_{R}{}_{\left(k\right)}\right)$$
where E and $E_{T}$ demonstrate the present energy level of the SN and the energy spent on data distributing, respectively.
(7)$$E_{T}\left(k,d\right)=kE_{e}+KE_{a}d^{2}$$
where $E_{e}$ stands for the electron energy, $E_{a}$ implies the energy has been utilized in implication, and $E_{R\left(k\right)}$ defines the energy transmitted on the received data.
(8)$$E_{R\left(k\right)}=kE_{e}$$

In addition, the mean value of the distance between neighboring nodes, which exist as single-hop neighbors, can be calculated as follows:(9)$$DIS=\frac{\sum_{j=1}^{NB_{i}}dist\left(i,nb_{j}\right)}{NB_{i}}$$

Although $dist\left(i,nb_{j}\right)$ describes distance of the SN from the neighboring *j*th SN, eventually, the NDEG implies the quantity of neighboring nodes, which have a transmitting radius, as follows:(10)NDEG=|N(x)|

At this point, $N\left(x\right)=\left\{\frac{n_{y}}{dist\left(x,y\right)}<trans_{range}\right\}x\neq y$, and dist(x,y) defines the distance between two nodes, $n_{x}$ and $n_{y}$, and $trans_{range}$ stands for the transmission range of the nodes.

### 3.3. Design of the CSRO Algorithm

The position of the lost human is the primary motivation of the search and rescue optimization method for optimization problems, and the significance of the clue originating in this position defines the cost of the solution. Now, the best method discloses a good position with additional hints [59]. When leaving certain clues, people seek the best option across the search method. However, the search position for the individual is kept in a situation matrix (matrix X), with the equivalent size of the memory matrix, and the left clue can be saved in a memory matrix (matrix M). n×d shows the problem parameter and n determines the individual quantity in the group.
(11)$$C=\left[\begin{array}{l} X\\ M \end{array}\right]=\left[\begin{array}{c} \begin{array}{ccc} X_{1,1} & \ldots & X_{1,d}\\ \vdots & \ddots & \vdots \\ X_{n,l} & \ldots & X_{n,d} \end{array}\\ \begin{array}{ccc} M_{1,1} & \cdots & M_{1,d}\\ \vdots & \ddots & \vdots \\ M_{n,l} & \cdots & M_{n,d} \end{array} \end{array}\right]$$

From the above equation, assuming there are random clues amongst the obtained clues, the search direction can be attained by the following:(12)$$sd_{i}=\left(X_{j}-C_{k}\right),\;\mathrm{Where}\;k\neq i$$
where $C_{k}$ denotes a random value among 1 and $2N,X_{i}$, and $C_{k}$ defines the position of the $i^{\mathrm{th}}$ human and the $k^{\mathrm{th}}$ clue, respectively. $sd_{i}$ indicates the search direction of the clues. It is noticed that $C_{k}$ equals $X_{j},$ k≠i. For avoiding repetitive position searches, the parameter of $X_{i}$ will not be modified by moving in the indicated direction [60,61,62,63,64]. The SAR approach uses a binomial crossover operator for applying to the limitations. Moreover, when the clue has greater significance compared with the current clue, a region was searched for the $sp_{i}$ direction. Otherwise, a search for the location of the existing position in the $sp_{i}$ is continuous. Therefore, the novel position of the $j^{\mathrm{th}}$ variable is expressed by the *i*th human, as follows:(13)$$X_{i,j}=\{\begin{array}{l} |C_{k,j}+r_{1}\times \left(X_{i,j}-C_{k,j}\right)\;if\;f\left(C_{k}\right)>f\left(X_{i}\right)\\ |X_{i,j}+r_{2}\times \left(X_{i,j}-C_{k,j}\right)\;if\;r_{2}<sp_{i}\;or\;j=j_{r},j-1,\ldots ,d\\ |X_{i}\;otherwise \end{array}$$
where $c_{k,j}$ denotes the position of variable j and the clue $k.\;j_{r},$ $r_{1}$, and $r_{2}$ represent three uniform random numbers within [1, d], [−1,1], and [0,1], respectively. The second stage is about the individual. Here, an exploitation term has been performed regarding the human’s current location [65,66,67,68]. This stage employs the distinct clues connection concept from the social stage. The position, upgraded by the human, i, can be attained as follows:(14)$$X_{i}=X_{j}+r_{3}\times \left(C_{k}-C_{m}\right)$$
where $r_{3}$ denotes a uniformly distributed number between 0 and 1, $C_{m}\;$and $C_{k}$ define two arbitrary numbers between 1 and 2 N, respectively, and i≠k≠m. They could testify that they are in the solution space afterward, solving the solution from the preceding stages. This phase is named the boundary. In such cases, the following formula is utilized when the solution is placed outside the border:(15)$$X_{ij}^{\prime }=\{\begin{array}{l} \frac{\left(X_{ij}+X_{j}^{\;\max\;}\right)}{2}\;if\;X_{ij}^{\prime }>X_{j}^{\;\max\;}\\ \frac{\left(X_{i,j}+X_{j}^{\;\min\;}\right)}{2}\;if\;X_{ij}^{\prime }<X_{j}^{\;\min\;} \end{array}\;$$
where j=1,2,…,d, $X_{j}^{\min}$, and $X_{j}^{\max}$ represent the minimal and maximal thresholds for the parameter j, respectively. According to this stage, the lost human candidate is searched for on the basis of the previously elucidated technique. When the sum of the cost function in a given scenario, $X_{i}^{\prime }\left(f\left(X_{i}^{\prime }\right)\right)$, is superior to the existing one, $\left(f\left(X_{i}\right)\right)$, then the preceding location (*X*) would be saved in an accidental position in the memory matrix (*M*), and would be described as novel situation. If not, then the situation would be left, and the memory would not be upgraded.
(16)$$M_{n}=\{\begin{array}{l} X_{j}\;if\;f\left(X_{i}^{\prime }\right)>f\left(X_{i}\right)\\ M_{n}\;otherwise \end{array}\;$$
(17)$$X_{i}=\{\begin{array}{l} X_{i}^{\prime }\;if\;f\left(X_{i}^{\prime }\right)>f\left(X_{i}\right)\\ X_{i}\;otherwise \end{array}\;$$
where n determines a random integer among 1 and N, and $M_{n}$ defines the location of clue number, n, in the memory matrix.

Time is critical in locating lost individuals due to threat of injury, and any delay that occurs at the time of searching could lead to death [69,70,71,72]. Hence, when a person does not discover a notable clue during their search, it leaves the next person with the existing position.
(18)$$usn_{i}=\{\begin{array}{l} usn_{i}+1\;if\;f\left(X_{i}^{\prime }\right)<f\left(X_{i}\right)\\ 0\;otherwise \end{array}\;$$
where usn determines the unproductive searching number. When usn is superior to MU for a person, it moves towards the other location in the space solution. Figure 2 portrays the flow chart of SRO method.

Search and rescue optimization (SRO), when searching for people, typically takes place in two distinct phases: social and individual. Collection followers search for clues founded in their location, and focus on areas that are more likely to yield clues in the social phase. There is no regard for where or how many clues have been found by others during the individual search phase. In general, clues fall into two categories, as follows:Remember to save a clue: A member of the exploration group is present and searching the surrounding area.Forgotten clue: Members of the group discovered the clue, but no one is in the location to solve it. To put it another way, the person who discovered the hint has abandoned it in search of further potentially relevant information. Members of the group, on the other hand, have access to the information about that clue.

To increase exploration competence and ensure that the ideal answer is reached, the chaotic approach is combined with the krill head algorithm (KHA). Because Chebyshev maps are the most utilized chaotic behavioral maps, chaotic sequences are likely to be created efficiently and fast. Furthermore, longer sequences are not necessary.

Therefore, the existing solution has been switched to an accidental resolution in the solution space, according to Equation (7) for a possible solution, when usn>MU (Multi−User). In addition, for an unfeasible solution, usn is superior to MU, the memory matrix solution using the minimum number of limitation violations is selected, and the current solution is switched through the solution, so that the current solution substitutes the memory matrix.
(19)$$X_{ij}=X_{j}^{\;\min\;}+r_{4}\times \left(X_{j}^{\;\max\;}-X_{j}^{\;\min\;}\right)\;j=1,2,\;d$$
where $r_{4}$ shows a frequently distributed random number within the range of 0–1. Under instruction to recover the worldwide optimization capability of an SRO algorithm, the chaotic concept is integrated into it. The chaotic state is an unstable state, which is extremely sensitive to initial conditions, which can be utilized for avoiding the local optimum problem and improving the quality of the solution. It is applied for achieving improved exploration and exploitation in every searching region, thereby enhancing the outcome in determining optimum global solutions [46,47,48]. The chaotic map was used in this study to indicate human searches around their current location in the individual phase, and the concept of linking different hints is used in the communal stage for exploration. In contrast to the social stage, the separation stage changed every dimension of $X_{ij}$. Time is an important factor in the search-and-rescue procedure, because missing individuals may be hurt, and search-and-rescue parties arriving late may result in death. As a result, these processes should be designed in such a way that a huge amount of data is examined in as short a period of time as possible. As a result, if a human did not find the best hints after running a specific search count in their location, they would leave and go to a different site.
(20)$$x_{i}^{k+1}=x_{i}^{k}+C_{nap}\times \left(x_{BH}-x_{i}^{k}\right),\;i=1,2,\ldots ,N$$
where $x_{i}^{k}$ and $x_{i}^{k+1}$ denote positions at iterations k and k+1 and $C_{map}$ represents a chaotic map. In this work, ten chaotic maps were used to determine the random values involved in the SRO algorithm.

### 3.4. Application of CSRO Algorithm for Data Transmission

The main function of the CSRO algorithm is maximizing the lifespan of networks and minimizing energy utilization of all sensor nodes. Assume that $h_{1}$ is the most objective function, such that CH is selected as the next hop, CH, with a superior RE, to route the data; such that, for maximizing the network lifespan, for instance, $h_{1}$ is maximization. Assume that $h_{2}$ is another main function that has a minimal distance among the CHs to the next hop CH, and the next hop CH, to the base station (BS). This procedure occurs under the instruction to decrease the energy utilization of the network as required for minimizing $h_{2}$. Assume that $h_{3}$ is the third main function; thus, CHs is selected as the next hop among the CHs with a lesser node degree. In order to improve the lifespan of the network, $h_{3}$ must be minimized. Assume that $b_{ij}$ is a Boolean variable, determined as follows:(21)$$b_{ij}=\{\begin{array}{cc} 1 & if\;next-hop\left(CH_{i}\right)=CH_{j},\;\forall_{i,j}1\leq i,\;j\leq m\;\\ 0 & Otherwise \end{array}$$
(22)$$\mathrm{Minimize}\;F=\frac{1}{h_{1}}\times \beta_{1}+h_{2}\times \beta_{2}+h_{3}\times \beta_{3}$$
which is subject to the following:(23)$$dis\left(CH_{i},CH_{j}\right)\leq d_{max}CH_{j}\left(C+BS\right)$$
(24)$$\overset{m}{\underset{j=1}{\sum }}b_{ij}=1\;and\;1\neq j$$

The constraint in (23) means that the next hop node of ${CH}_{i}$ will be in the range of ${CH}_{i}$, and that the next hop node is ${CH}_{j}$. $\beta_{1,\;}\;\beta_{2},\beta_{3}$ indicate the anchor nodes with the target distance. The constraint in (25) declares that the next hop node of ${CH}_{i}$ is unique, for instance, ${CH}_{j}$, and the constraint makes sure that there could not be 0 or 100% weight on either of the objectives.

## 4. Performance Validation

This section analyzes the CSRO-MHDT method in comparison with recent methods of effective data transmission processes in a UWSN. The consequences are reviewed under variable rounds of execution. Table 2 and Figure 3 demonstrate an examination of the comparative number of alive nodes (NAN) of the CSRO-MHDT technique under varying rounds. The results indicated that the CSRO-MHDT technique has increased values of NAN under all rounds. For instance, with 440 rounds, the CSRO-MHDT technique has obtained a higher NAN of 300, whereas the energy-efficient clustering routing protocol (EECRP), the fuzzy C-means and moth–flame optimization (FCMMFO), the fuzzy scheme and particle swarm optimization (FBCPSO), the energy-efficient grid routing based on 3D cubes (EGRC), and the low-energy adaptive clustering hierarchy based on expected residual energy (LEACH-ERE) models have attained lower NAN results of 296, 290, 299, 298, and 287 nodes, respectively. In addition, with 600 rounds, the CSRO-MHDT technique has an increased NAN result of 279, whereas the EECRP, FCMMFO, FBCPSO, EGRC, and LEACH-ERE models have lower NAN results of 264, 253, 240, 243, and 238 nodes, respectively. Moreover, with 920 rounds, the CSRO-MHDT technique accomplished a higher NAN result of 68, whereas the EECRP, FCMMFO, FBCPSO, EGRC, and LEACH-ERE models demonstrated lower NAN results of 20, 0, 0, 0, and 0 nodes, respectively.

First node dies (FND), half node dies (HND), and last node dies (LND) are examples of the comparative network area evaluation of the CSRO-MHDT technique, as shown in Table 3 and Figure 4, respectively (LND). The results indicated that the CSRO-MHDT technique has resulted in lengthened lifetime over the existing methods. With respect to FND, the CSRO-MHDT method reached FND at 476 rounds, whereas the EECRP, FCMMFO, FBCPSO, EGRC, and LEACH-ERE models attained FND at earlier rounds of 440, 403, 361, 323, and 280, respectively. In addition, in terms of HND, the CSRO-MHDT system reached HND at 838 rounds, whereas the EECRP, FCMMFO, FBCPSO, EGRC, and LEACH-ERE techniques reached HND at earlier rounds of 754, 749, 730, 753, and 722, respectively. Finally, with respect to LND, the CSRO-MHDT approach reached LND at 998 rounds, whereas the EECRP, FCMMFO, FBCPSO, EGRC, and LEACH-ERE techniques reached LND at earlier rounds of 940, 921, 903, 874, and 840, respectively. 

Next, a brief TEC investigation of the CSRO-MHDT method in comparison with existing approaches is provided in Table 4 and Figure 5. The results indicated that the CSRO-MHDT technique had the lowest TEC under all rounds compared with existing methods. For example, with 50 rounds, the CSRO-MHDT method obtained lower TEC of 2.22%, whereas the EECRP, FCMMFO, FBCPSO, EGRC, and LEACH-ERE models achieved higher TEC of 2.46%, 2.71%, 5.39%, 5.15%, and 7.59%, respectively. With 500 rounds, the CSRO-MHDT system had the lowest TEC of 34.16%, whereas the EECRP, FCMMFO, FBCPSO, EGRC, and LEACH-ERE models achieved higher TEC of 39.29%, 39.29%, 56.84%, 46.60%, and 65.87%, respectively. Finally, with 1000 rounds, the CSRO-MHDT technique had the lowest TEC of 99%, whereas the EECRP, FCMMFO, FBCPSO, EGRC, and LEACH-ERE methodologies attained higher TEC of 99.55%, 99.52%, 99.61%, 100%, and 100%, respectively.

A detailed PLR analysis of the CSRO-MHDT approach in comparison with recent methods is offered in Table 5 and Figure 6. The outcomes showed that the CSRO-MHDT technique had the lowest PLR under all rounds compared with existing approaches. For instance, with 50 rounds, the CSRO-MHDT algorithm had decreased PLR of 1%, whereas the EECRP, FCMMFO, FBCPSO, EGRC, and LEACH-ERE systems obtained higher PLR of 1%, 1%, 1%, 1%, and 1%, respectively. Next, with 500 rounds, the CSRO-MHDT technique had a low PLR of 2%, whereas the EECRP, FCMMFO, FBCPSO, EGRC, and LEACH-ERE models had higher PLR results of 5%, 7%, 8%, 9%, and 10%, respectively. Finally, with 1000 rounds, the CSRO-MHDT technique had a low PLR of 12%, whereas the EECRP, FCMMFO, FBCPSO, EGRC, and LEACH-ERE models had increased PLR of 17%, 19%, 22%, 23%, and 25%, respectively.

Table 6 and Figure 7 illustrate the number of packets received (NPR) by the CSRO-MHDT system under varying rounds, in comparison with existing approaches. The results showed that the CSRO-MHDT technique had higher values of NPR under all rounds. For instance, with 50 rounds, the CSRO-MHDT technique attained higher NPR of 3792%, whereas the EECRP, FCMMFO, FBCPSO, EGRC, and LEACH-ERE approaches had lower NPR results of 3400%, 2549%, 2876%, 2026%, and 1554%, respectively. With 500 rounds, the CSRO-MHDT method had higher NPR of 17,924%, whereas the EECRP, FCMMFO, FBCPSO, EGRC, and LEACH-ERE models had lower NPR results of 17,401%, 16,027%, 14,784%, 12,167%, and 11,837%, respectively. Furthermore, with 1000 rounds, the CSRO-MHDT system accomplished a higher NPR result of 23,098%, whereas the EECRP, FCMMFO, FBCPSO, EGRC, and LEACH-ERE algorithms demonstrated lower NPR results of 21,457%, 19,887%, 18,775%, 14,849%, and 14,427%, respectively.

Table 7 and Figure 8 depict the PDR analysis of the CSRO-MHDT algorithm under varying rounds in comparison with existing approaches. The outcomes revealed that the CSRO-MHDT technique resulted in increased values of PDR under all rounds. For instance, with 150 rounds, the CSRO-MHDT technique attained superior PDR of 99%, whereas the EECRP, FCMMFO, FBCPSO, EGRC, and LEACH-ERE models reached lower PDR results of 99%, 99%, 98%, 97%, and 97%, respectively. Moreover, with 500 rounds, the CSRO-MHDT approach had an increased PDR of 98%, whereas the EECRP, FCMMFO, FBCPSO, EGRC, and LEACH-ERE models obtained lower PDR results of 95%, 93%, 92%, 91%, and 90%, respectively. Finally, with 1000 rounds, the CSRO-MHDT technique accomplished a higher PDR of 88%, whereas the EECRP, FCMMFO, FBCPSO, EGRC, and LEACH-ERE techniques exhibited reduced PDR of 83%, 81%, 78%, 77%, and 75%, respectively.

By examining the above results and discussion, it can be confirmed that the CSRO-MHDT technique can accomplish effective data transmission in a UWSN.

## 5. Conclusions

In this study, a novel CSRO-MHDT method was developed for the optimal choice of routes for data transmission in a UWSN. The primary intention of the CSRO-MHDT technique is to reduce energy dissipation and enhance the lifetime of the UWSN. Primarily, the CSRO-MHDT technique involves WCA for effective choice of *CH*s. In addition, the CSRO-MHDT technique derived a fitness function and effectively selected a set of routes in a UWSN. For assessing the outcomes of the CSRO-MHDT technique, a wide-ranging experimental examination was carried out and the results were assessed under several aspects. The extensive comparative analysis highlighted the superior performance of the CSRO-MHDT technique over recent state-of-the-art approaches. Therefore, the CSRO-MHDT method can be used in application for optimal data transmission in UWSNs. In the future, delay-aware data aggregation schemes can be designed to improve the efficiency of UWSNs.

## Figures and Tables

**Figure 1 sensors-22-02867-f001:**
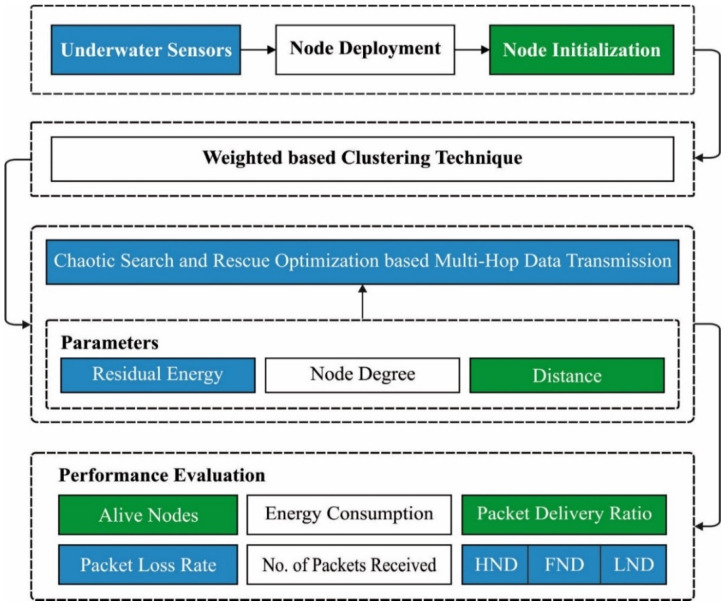
Overall process of the CSRO-MHDT technique.

**Figure 2 sensors-22-02867-f002:**
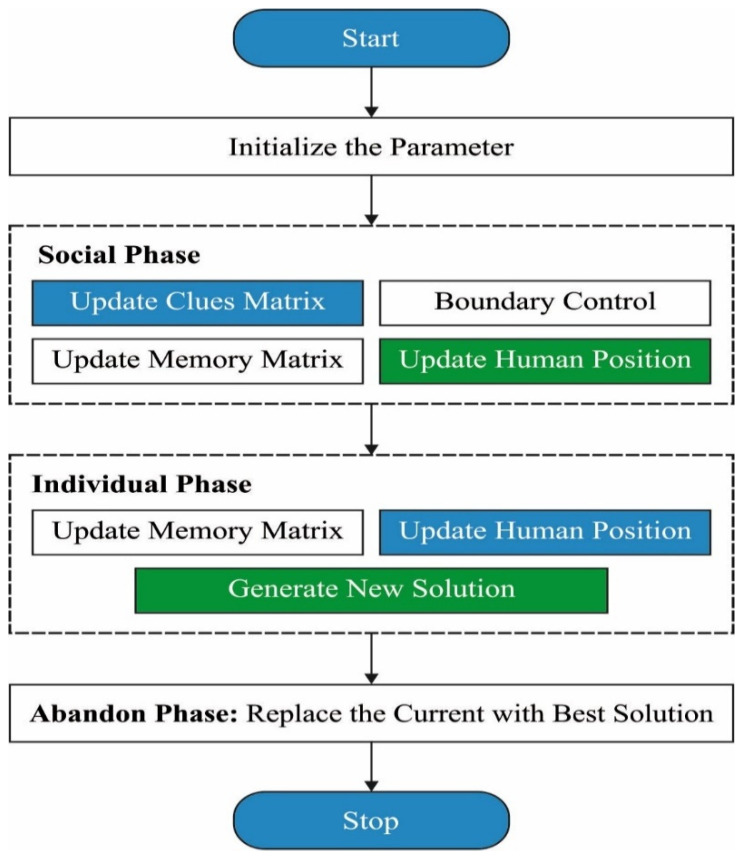
Flowchart of SRO.

**Figure 3 sensors-22-02867-f003:**
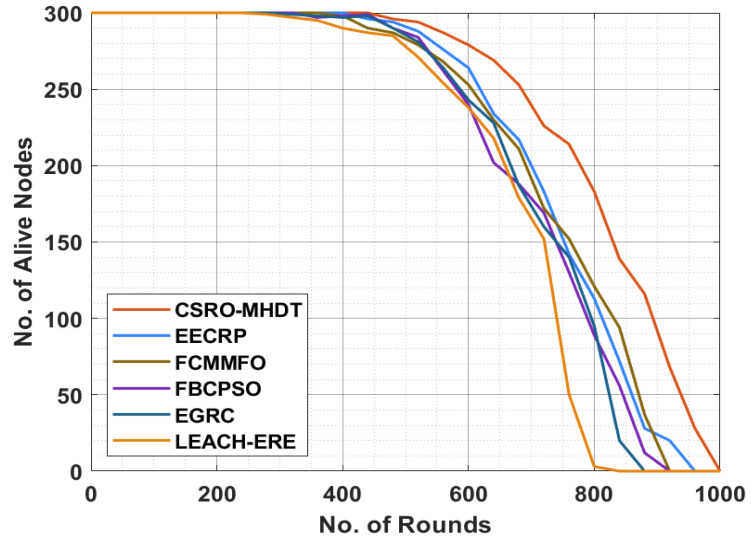
NAN analysis of the CSRO-MHDT technique in comparison with existing approaches.

**Figure 4 sensors-22-02867-f004:**
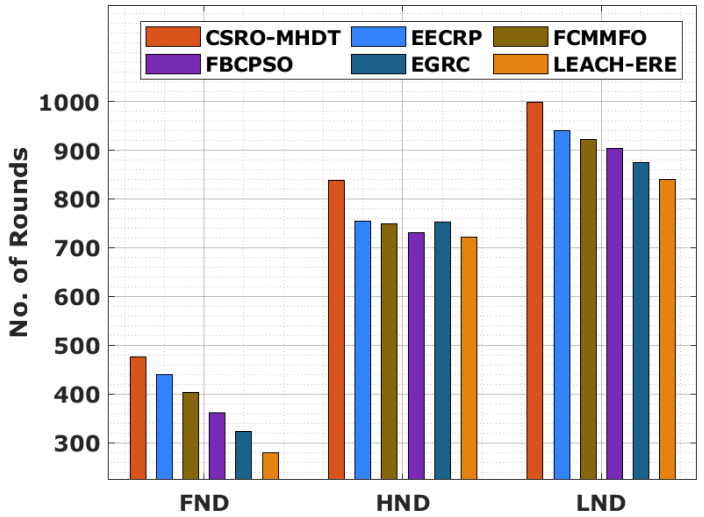
Network lifetime analysis of the CSRO-MHDT technique in comparison with existing approaches.

**Figure 5 sensors-22-02867-f005:**
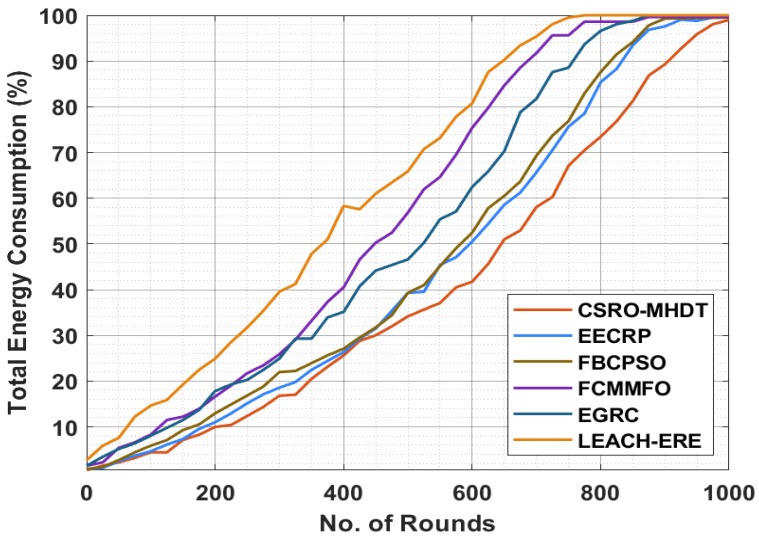
TEC analysis of the CSRO-MHDT method in comparison with current tactics.

**Figure 6 sensors-22-02867-f006:**
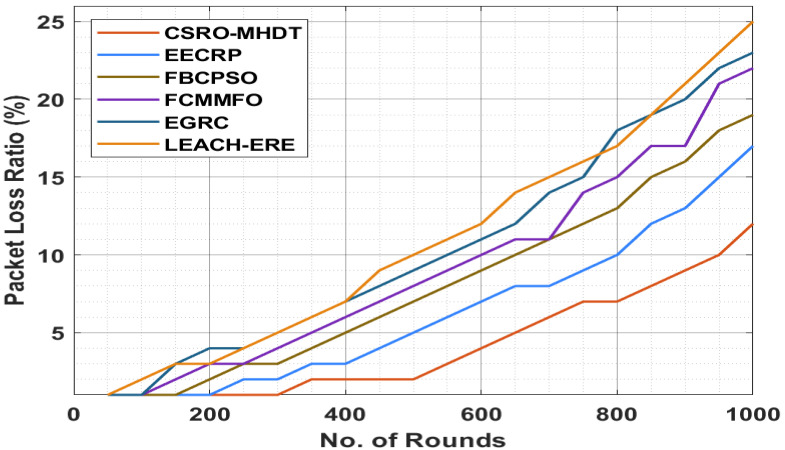
PLR analysis of the CSRO-MHDT technique in comparison with existing approaches.

**Figure 7 sensors-22-02867-f007:**
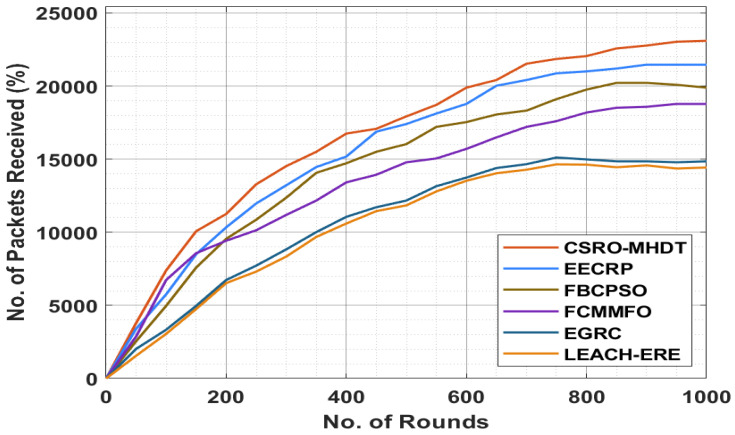
NPR analysis of the CSRO-MHDT technique in comparison with existing approaches.

**Figure 8 sensors-22-02867-f008:**
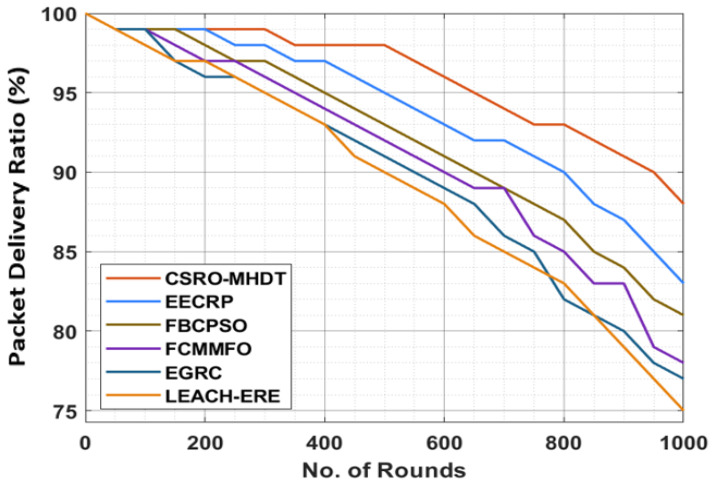
PDR analysis of the CSRO-MHDT technique in comparison with existing approaches.

**Table 1 sensors-22-02867-t001:** Summary of existing approaches.

Reference No.	Published Year	Approach	Advantages	Disadvantages
[26]	2019	Adaptive node clustering	Throughput and reliability are high in clustering process	Network life time
[27]	2020	LOCAN methodology	Energy consumption	Packet loss
[28]	2020	Bee algorithm	Reliable transmission	Not suitable for deep UWSNs
[29]	2021	Grey wolf optimization algorithm	Reducing packet loss and traffic systems	Sensor failure
[30]	2019	Threshold and energy level partition	Reducing delay and high throughput	Maximizing energy Resources
[36]	2021	Path selection strategy	High throughput	Time delay
[37]	2020	Fuzzy C means and moth–flame optimization method	Reducing energy consumption	Time delay Packet loss
Proposed Approach	2022	Chaotic search and rescue optimization	Reliable transmission, Reducing time delay and high throughput	Packet loss in high level nodes

**Table 2 sensors-22-02867-t002:** NAN study of the CSRO-MHDT technique in comparison with existing methods analyzed under different rounds.

No. of Alive Nodes						
No. of Rounds	CSRO-MHDT	EECRP	FCMMFO	FBCPSO	EGRC	LEACH-ERE
0	300	300	300	300	300	300
40	300	300	300	300	300	300
80	300	300	300	300	300	300
120	300	300	300	300	300	300
160	300	300	300	300	300	300
200	300	300	300	300	300	300
240	300	300	300	300	300	300
280	300	300	300	300	300	299
320	300	300	300	300	299	297
360	300	300	300	297	298	295
400	300	300	298	298	297	290
440	300	296	290	299	298	287
480	296	294	287	290	290	285
520	294	288	279	284	281	271
560	287	276	268	262	264	254
600	279	264	253	240	243	238
640	269	234	230	202	228	218
680	253	217	211	188	187	179
720	226	183	172	169	160	152
760	214	142	132	130	140	50
800	183	113	121	89	95	3
840	139	72	94	56	20	0
880	116	28	37	12	0	0
920	68	20	0	0	0	0
960	28	0	0	0	0	0
1000	0	0	0	0	0	0

**Table 3 sensors-22-02867-t003:** Network lifetime analysis of the CSRO-MHDT technique in comparison with existing approaches.

No. of Rounds						
	CSRO-MHDT	EECRP	FCMMFO	FBCPSO	EGRC	LEACH-ERE
FND	476	440	403	361	323	280
HND	838	754	749	730	753	722
LND	998	940	921	903	874	840

**Table 4 sensors-22-02867-t004:** TEC analysis of the CSRO-MHDT technique in comparison with existing approaches with different rounds.

Total Energy Consumption (%)						
No. of Rounds	CSRO-MHDT	EECRP	FBCPSO	FCMMFO	EGRC	LEACH-ERE
0	0.51	0.76	0.76	1.49	1.49	2.71
25	1.49	1.00	1.25	2.22	3.44	5.88
50	2.22	2.46	2.71	5.39	5.15	7.59
75	3.20	3.68	4.42	6.61	6.37	12.22
100	4.43	4.66	5.88	8.32	8.07	14.66
125	4.42	6.12	7.10	11.49	9.78	15.88
150	7.14	7.34	9.29	12.22	11.49	19.29
175	8.27	9.54	10.51	13.93	13.68	22.46
200	9.97	11.00	12.95	16.61	17.83	24.90
225	10.41	12.95	14.90	19.05	19.29	28.56
250	12.39	15.14	16.85	21.73	20.27	31.73
275	14.34	17.10	18.80	23.44	22.46	35.38
300	16.80	18.56	21.97	25.87	24.90	39.53
325	17.02	19.78	22.22	29.04	29.29	41.24
350	20.48	22.46	23.92	33.19	29.29	47.82
375	23.14	24.41	25.63	37.33	33.92	50.99
400	25.61	26.36	27.09	40.50	35.14	58.31
425	28.80	29.29	29.53	46.60	40.75	57.57
450	30.02	31.48	31.73	50.26	44.16	60.99
475	31.97	35.38	34.41	52.45	45.38	63.43
500	34.16	39.29	39.29	56.84	46.60	65.87
525	35.63	39.53	40.99	61.96	50.26	70.74
550	37.09	45.38	45.14	64.65	55.38	73.18
575	40.50	47.09	49.04	69.52	57.09	77.81
600	41.72	50.50	52.45	75.38	62.45	80.74
625	45.63	54.40	57.82	79.76	65.87	87.57
650	50.99	58.55	60.50	84.64	70.25	90.25
675	52.94	61.23	63.67	88.54	78.79	93.42
700	58.06	65.62	69.28	91.71	81.72	95.37
725	60.26	70.50	73.67	95.61	87.57	98.05
750	67.08	75.62	76.84	95.61	88.54	99.52
775	70.50	78.55	82.93	98.61	93.66	100.00
800	73.42	85.37	87.57	98.61	96.59	100.00
825	76.84	88.30	91.47	98.61	98.05	100.00
850	81.23	93.42	94.15	98.61	98.78	100.00
875	86.84	96.83	97.81	99.61	100.00	100.00
900	89.27	97.57	99.27	99.61	100.00	100.00
925	92.69	99.03	99.34	99.61	100.00	100.00
950	95.86	98.78	99.43	99.61	100.00	100.00
975	98.05	99.52	99.45	99.61	100.00	100.00
1000	99.00	99.55	99.52	99.61	100.00	100.00

**Table 5 sensors-22-02867-t005:** PLR analysis of the CSRO-MHDT technique in comparison with existing approaches.

Packet Loss Ratio (%)						
No. of Rounds	CSRO-MHDT	EECRP	FBCPSO	FCMMFO	EGRC	LEACH-ERE
0	0	0	0	0	0	0
50	1	1	1	1	1	1
100	1	1	1	1	1	2
150	1	1	1	2	3	3
200	1	1	2	3	4	3
250	1	2	3	3	4	4
300	1	2	3	4	5	5
350	2	3	4	5	6	6
400	2	3	5	6	7	7
450	2	4	6	7	8	9
500	2	5	7	8	9	10
550	3	6	8	9	10	11
600	4	7	9	10	11	12
650	5	8	10	11	12	14
700	6	8	11	11	14	15
750	7	9	12	14	15	16
800	7	10	13	15	18	17
850	8	12	15	17	19	19
900	9	13	16	17	20	21
950	10	15	18	21	22	23
1000	12	17	19	22	23	25

**Table 6 sensors-22-02867-t006:** NPR analysis of the CSRO-MHDT technique in comparison with existing approaches.

No. of Packets Received (%)						
No. of Rounds	CSRO-MHDT	EECRP	FBCPSO	FCMMFO	EGRC	LEACH-ERE
0	0	0	0	0	0	0
50	3792	3400	2549	2876	2026	1554
100	7391	5755	4970	6736	3334	3054
150	10,073	8503	7587	8568	4970	4751
200	11,251	10,335	9550	9419	6736	6521
250	13,279	11,970	10,858	10,138	7718	7310
300	14,522	13,213	12,363	11,185	8830	8336
350	15,503	14,457	14,064	12,167	10,008	9679
400	16,746	15,176	14,718	13,410	11,054	10,595
450	17,074	16,877	15,503	13,933	11,709	11,442
500	17,924	17,401	16,027	14,784	12,167	11,837
550	18,709	18,120	17,204	15,045	13,148	12,789
600	19,887	18,775	17,532	15,700	13,737	13,521
650	20,410	20,018	18,055	16,485	14,391	14,034
700	21,523	20,410	18,317	17,204	14,653	14,276
750	21,850	20,868	19,102	17,597	15,111	14,642
800	22,046	20,999	19,756	18,186	14,980	14,623
850	22,569	21,195	20,214	18,513	14,849	14,445
900	22,766	21,457	20,214	18,578	14,849	14,570
950	23,027	21,457	20,083	18,775	14,784	14,358
1000	23,093	21,457	19,887	18,775	14,849	14,427

**Table 7 sensors-22-02867-t007:** PDR analysis of the CSRO-MHDT technique in comparison with existing approaches.

Packet Delivery Ratio (%)						
No. of Rounds	CSRO-MHDT	EECRP	FBCPSO	FCMMFO	EGRC	LEACH-ERE
0	100	100	100	100	100	100
50	99	99	99	99	99	99
100	99	99	99	99	99	98
150	99	99	99	98	97	97
200	99	99	98	97	96	97
250	99	98	97	97	96	96
300	99	98	97	96	95	95
350	98	97	96	95	94	94
400	98	97	95	94	93	93
450	98	96	94	93	92	91
500	98	95	93	92	91	90
550	97	94	92	91	90	89
600	96	93	91	90	89	88
650	95	92	90	89	88	86
700	94	92	89	89	86	85
750	93	91	88	86	85	84
800	93	90	87	85	82	83
850	92	88	85	83	81	81
900	91	87	84	83	80	79
950	90	85	82	79	78	77
1000	88	83	81	78	77	75

## Data Availability

Not applicable.

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
