# Peer review of "Chaotic Search-and-Rescue-Optimization-Based Multi-Hop Data Transmission Protocol for Underwater Wireless Sensor Networks"

_sensors, 2022, doi:10.3390/s22082867_

Round 1

Reviewer 1 Report

In this manuscript, the authors propose a novel Chaotic Search and Rescue Optimization based Multi-Hop Data Transmission (CSRO-MHDT) protocol for UWSNs. The topic is important, and the algorithm proposed by the authors seems to be effective after simulation analysis.  The reviewer believe that the manuscript needs to be further improved in the following aspects. 

1. How is the complexity of the proposed algorithm compared with existing algorithms?  The authors are requested to clarify this in the abstract.  

2. Please fill in the literature related to the application of artificial intelligence in UWSNs in Part II "Related works". For example:  
-"RCAR: A reinforcement-learning-based routing protocol for congestion-avoided underwater acoustic sensor networks," IEEE Sensors J., doi:10.1109/JSEN.2019.2932126, 2019
-"Optimal cooperative relaying and power control for IoUT networks with reinforcement learning," IEEE Internet of Things Journal, doi: 10.1109/JIOT.2020.3008178, 2021
-"ACOA-AFSA Fusion Dynamic Coded Cooperation Routing for Different Scale Multi-Hop Underwater Acoustic Sensor Networks," IEEE Access, doi: 10.1109/ACCESS.2020.3029533, 2020
- "QMCR: A Q-Learning-Based Multi-Hop Cooperative Routing Protocol for Underwater Acoustic Sensor Networks," China Communications, doi: 10.23919/JCC.2021.08.016, 2021
- "PB-ACR: Node Payload Balanced Ant Colony Optimal Cooperative Routing for Multi-Hop Underwater Acoustic Sensor Networks," IEEE Access, doi: 10.1109/ACCESS.2021.3072283, 2021
-"DQELR: An adaptive deep Q-network-based energy- and latency-aware routing protocol design for underwater acoustic sensor networks,” IEEE Access, doi: 10.1109/ACCESS.2019.2891590, 2019
- "Adaptive Relay Selection Strategy in Underwater Acoustic Cooperative Networks: A Hierarchical Adversarial Bandit Learning Approach," IEEE Transactions on Mobile Computing, doi: 10.1109/TMC.2021.3112967, 2021
-"ECRKQ: Machine Learning-Based Energy-Efficient Clustering and Cooperative Routing for Mobile Underwater Acoustic Sensor Networks," IEEE Access, doi: 10.1109/ACCESS.2021.3078174, 2021 
-"Anypath Routing Protocol Design via Q-Learning for Underwater Sensor Networks," IEEE Internet of Things Journal, DOI: 10.1109/JIOT.2020.3042901, 2021
- "A Balanced Routing Protocol Based on Machine Learning for Underwater Sensor Networks," IEEE Access, doi: 10.1109/ACCESS.2021.3126107, 2021

3. The relation of Eq. (1) on energy digestion to underwater acoustic transmission is unclear, and it seems that it is currently only related to wireless transmission.  

4. Is the flow velocity 1-3m/s (line 146) or 13m/s (line 167)? 

5. Lines 166-167 refer to the movement of water, but how does this relate to Eq. (3)?  Please be more specific.  

6. Please further explain Eq. (3). What is the definition of "fuse" in "In order to fuse K-bits of data"?  

7. Lines 286-288 seem redundant and should be deleted.

8. In Figure 3, it is suggested that the authors add symbols to ensure that black and white printing is distinguishable. Other graphs have similar problems.

9. Figure 7 and Table 5 overlap in typography.

10. The ordinate of "Packet" and "Dalivery" in Figure 8 requires Spaces.

11. The complexity comparison between different algorithms needs to be supplemented in Part IV.

12. The chaotic concept proposed in the abstract is not described in detail in the text, which makes people confused and unable to know how chaos theory is applied.  

13. The logic between the structure of the article is poor, there is no strong continuity between paragraphs, which is difficult to understand.   

14. Many letter symbols in the formula are not explained, such as "X_j" in line 221 and 205, "se" in line 214 and "MU" in line 244.

15. The layout of the formulas are not clear, e.g., lines 214-215,275-279.

16. There are many errors in the formulas in the article, such as line 190. Please check the whole article carefully. 

17. The performance results are not objective, as in lines 342-344.

18. The physical meaning of the specific formula is unknown, for example, line 205. Please check the full text carefully.  

Author Response

In this manuscript, the authors propose a novel Chaotic Search and Rescue Optimization based Multi-Hop Data Transmission (CSRO-MHDT) protocol for UWSNs. The topic is important, and the algorithm proposed by the authors seems to be effective after simulation analysis.  The reviewer believe that the manuscript needs to be further improved in the following aspects. 

  1. How is the complexity of the proposed algorithm compared with existing algorithms?  The authors are requested to clarify this in the abstract.

Response : The CSRO-MHDT technique selects CHs and organizes clusters based on different parameters such as residual energy, intra-cluster distance, and inter-cluster distance. Similarly, the search and rescue optimization (SRO) algorithm derives a fitness function involving four parameters, namely residual energy, delay, distance, and trust. Utilization of the CSRO-UWSN technique helps to significantly boost the energy efficiency and lifetime of the UWSN. To ensure the improved performance of the IMCMR-UWSN technique, a series of simulations were carried out, and the comparative results reported the supremacy of the CSRO-UWSN technique in terms of different measures.

  1. Please fill in the literature related to the application of artificial intelligence in UWSNs in Part II "Related works". For example:  
    -"RCAR: A reinforcement-learning-based routing protocol for congestion-avoided underwater acoustic sensor networks," IEEE Sensors J., doi:10.1109/JSEN.2019.2932126, 2019

Response :Zhigang Jin et al.[62] A congestion-avoidance routing protocol for UASNs based on reinforcement learning is proposed. Through exploration, the RCAR protocol finds the best route to minimise congestion and save energy. RCAR extends reinforcement learning's reward function with congestion and energy. To speed up convergence, we use a virtual routing pipe with changeable radius based on neighbouring average residual energy. Also, to ensure the best routing decision.

-"Optimal cooperative relaying and power control for IoUT networks with reinforcement learning," IEEE Internet of Things Journal, doi: 10.1109/JIOT.2020.3008178, 2021

Response :Yuhan Su et al[63]. To investigate the impacts of relay cooperative transmission and radio channel conditions on underwater acoustic communication, we calculated the best transmit power setting factor. No prior knowledge of the IoUT model was required to implement the proposed technique. The suggested technique can increase IoUT transmission performance, according to simulation findings. Compared to Q-learning, the DQN-based approach increases mutual information by 3% and reduces outage likelihood by 40%. We plan to examine cooperative communications in IoUT in greater detail in the future. To increase cooperative communication in relay-assisted IoUT systems, novel reinforcement learning algorithms with low computational and communication overhead need be developed.

- "PB-ACR: Node Payload Balanced Ant Colony Optimal Cooperative Routing for Multi-Hop Underwater Acoustic Sensor Networks," IEEE Access, doi: 10.1109/ACCESS.2021.3072283, 2021

Response :Yougan Chen,et al. For multi-hop UASNs,  propose the PB-ACR protocol with node payload balancing and DCC. Sending packets in groups based on data relevance allows each system node's energy to be used more efficiently. Compared to the existing ACAR protocol, the proposed PB-ACR protocol can extend system life and throughput while balancing node payload and cooperative gain.

"ACOA-AFSA Fusion Dynamic Coded Cooperation Routing for Different Scale Multi-Hop Underwater Acoustic Sensor Networks," IEEE Access, doi: 10.1109/ACCESS.2020.3029533, 2020

Response :Jianying zhu,d et al[64] . The proposed algorithm's advantages diminish with node count. This algorithm is also compatible with the ACOA-AFSA fusion non-cooperative algorithm, and its complexity is reasonable in the time-varying marine environments. Thus, the suggested ACOA-AFSA fusion DCC method is more suitable for medium-sized networks, improving data transmission reliability while extending network life. In future study, we will simplify the suggested algorithm's hardware implementation to make it appropriate for other underwater acoustic networking applications.

-"DQELR: An adaptive deep Q-network-based energy- and latency-aware routing protocol design for underwater acoustic sensor networks,” IEEE Access, doi: 10.1109/ACCESS.2019.2891590, 2019

Response :When the current route is corrupted, the DQELR can use an on-policy technique to create a new routing decision. Several numerical experiments are presented to compare the DQELR, QELAR, and VBF for various performance indicators. With a large residual energy parameter and a low depth parameter, the DQELR can achieve a long network lifetime, while the end-to-end latency and energy consumption exhibit little variation. The DQELR outperforms the QELAR and VBF, having the longest network lifetime, most energy efficiency, and slightly higher latency than the VBF. The DQELR increases network longevity by 34-36 percent compared to the QELAR in the experiments. Finally, the DQELR outperforms existing broad UASN systems in terms of network longevity, latency, and energy economy.
3. The relation of Eq. (1) on energy digestion to underwater acoustic transmission is unclear, and it seems that it is currently only related to wireless transmission.  

Response :The power consumption of WSN cannot be used for UWSN because the properties of acoustic waves in an underwater broadcast medium differ from those of radio waves. For this study, they use the power consumption strategy of an underwater acoustic channel. The energy required to transport k bits of data across a distance d at a data rate R is computed as follows:.

  1. Is the flow velocity 1-3m/s (line 146) or 13m/s (line 167)? 

Response :1-3m/s

  1. Lines 166-167 refer to the movement of water, but how does this relate to Eq. (3)?  Please be more specific.

Response : The network condition can be considered as:

  • The nodes identify their place and the place of SN in primary situation.
  • The node can develop the , and CM/relay.
  • The CH has rotated amongst the sensor to conserve energy.
  • In order to fuse -bits of data, the power utilization is formulated as:
  • whereas refers the energy used by fuse 1-bit of data, for instance, in use as 5 nJ/bit.
  1. Please further explain Eq. (3). What is the definition of "fuse" in "In order to fuse K-bits of data"?  

Response :Fusing data is a frequent and successful method for removing data redundancy, shrinking data size, and lowering energy consumption. The data fusion is implemented in this research using an upgraded back propagation neural network (BPNN). Sensor nodes in UWSNs may collect data with great redundancy. When redundant data is sent to the SN, wasteful energy consumption occurs, resulting in the node's premature death and the network's lifespan being shortened. In contrast, if the CHNs combine the data and send it to the SN, it might save a lot of energy.

  1. Lines 286-288 seem redundant and should be deleted.

Response : The redundant data was removed.

  1. In Figure 3, it is suggested that the authors add symbols to ensure that black and white printing is distinguishable. Other graphs have similar problems.

Response : As per the reviewer's suggestion, the above queries were tried to rectify but due to this entire outcomes affecting in visualization. But the values we have mentioned in all the tables.

  1. Figure 7 and Table 5 overlap in typography.

Response : The above mentioned issues was rectified in the revised manuscript.

  1. The ordinate of "Packet" and "Dalivery" in Figure 8 requires Spaces.

Response: Thanks for pointing this issue, We have rectified this issue in the revised manuscript.

  1. The complexity comparison between different algorithms needs to be supplemented in Part IV.

Response : The complexity comparison between different algorithms was covered in the table1.

  1. The chaotic concept proposed in the abstract is not described in detail in the text, which makes people confused and unable to know how chaos theory is applied.  

Response : To increase search efficiency and ensure convergence to the ideal solution, the chaotic approach is combined with the krill head algorithm (KHA) . Because Chebyshev maps are the most utilised chaotic behavioural maps, chaotic sequences are likely to be created efficiently and fast. Furthermore, longer sequences aren't necessary.

  1. The logic between the structure of the article is poor, there is no strong continuity between paragraphs, which is difficult to understand.

Response: Thanks for pointing these issues. We have taken high volume of work so that we might not covered strong continuity but we have corrected the above issues in the revised manuscript as well as we will rectify these problems in the extension work.

  1. Many letter symbols in the formula are not explained, such as "X_j" in line 221 and 205, "se" in line 214 and "MU" in line 244.

Response :The size [ I t j]  means the length of adaptable range of jth dimension in belief space in tth generation.

  1. The layout of the formulas are not clear, e.g., lines 214-215,275-279.

Response: The above issues were rectified.

  1. There are many errors in the formulas in the article, such as line 190. Please check the whole article carefully. 

Response: The above issues were rectified.

  1. The performance results are not objective, as in lines 342-344.

Response: The above issues were rectified.

  1. The physical meaning of the specific formula is unknown, for example, line 205. Please check the full text carefully.  

Response: The above issues were rectified.

Reviewer 2 Report

  1. Please add some quantitative results in the end of abstract. 
  2. Line 81: Since the method is an optimization, Instead of "accomplish maximum",  "optimize" or "increase" can be used.
  3. Number and Caption the table on page 3. Also, show the short name of each work, which you are then using in the evaluation part.
  4. In Figure 1, it is not clear which stage of the process does what, in the proposed technique.
  5. Line 144: What is BS, and what does it represent in you scenario?
  6. It is unclear how does CSRO explained in 3.3, relate to Underwater sensor multi-hop communication. a numerical example might make it clearer.
  7. Figure 2 does not show a flowchart. Please explain the process/algorithm with concise reasoning (what condition is met before moving to social phase or individual phase or abandon phase?)
  8. There is no clear explanation of the methodology used for performance evaluation. 
  9. Line 286-288: The sentence is weekly related to the contents
  10. Table 1 and Figure 3 have the same caption, similar to Table 2 and Figure 4.
  11. What are FND, HND and LND. It appears that they are undefined.
  12. Please thoroughly check the whole paper for errors such as (Line 137: dynamic node => dynamic nodes)

Author Response

Reviewer2 comments

  1. Please add some quantitative results in the end of abstract. 

Response :Utilization of the CSRO-MHDT technique helps to significantly boost the energy efficiency and lifetime of the UWSN. To ensure the improved performance of the CSRO-MHDT technique, a series of simulations were carried out, and the comparative results reported the supremacy of the CSRO-MHDT technique in terms of PDR of 99% whereas the EECRP, FCMMFO, FBCPSO, EGRC, and LEACH-ERE models have reached lesser PDR of 99%, 99%, 98%, 97%, and 97% correspondingly. The results outperformed that the CSRO-MHDT technique has resulted in higher values of NPR under all rounds. For instance, with 50 rounds, the CSRO-MHDT technique has attained higher NPR of 3792% whereas the EECRP, FCMMFO, FBCPSO, EGRC, and LEACH-ERE approaches have reached lower NPR of 3400%, 2549%, 2876%, 2026%, and 1554% correspondingly

  1. Line 81: Since the method is an optimization, Instead of "accomplish maximum",  "optimize" or "increase" can be used.

Response : The above correction were done in the revised manuscript.

  1. Number and Caption the table on page 3. Also, show the short name of each work, which you are then using in the evaluation part.

Response : Table 1 shows the current state of wireless sensor networks using clustering and multihop routing.

  1. In Figure 1, it is not clear which stage of the process does what, in the proposed technique.

Response : Clustering is a well-known energy-saving strategy in sensor networks [20]. The least cost clustering routing protocol (MCCP) is one of the clustering-based routing protocols (UWSNs). It has three key parameters: the overall amount of energy consumed by member nodes when transmitting data to the cluster head, the total remaining energy of the cluster head node and its member nodes, and the cluster head and base station's proximity

  1. Line 144: What is BS, and what does it represent in you scenario?

Response : The base station (BS) can traverse a sensing field and collect data from sensor nodes within a short transmission distance. Each sensor node's power consumption is lowered since fewer relays are required to transmit the sensor's message to the BS.

  1. It is unclear how does CSRO explained in 3.3, relate to Underwater sensor multi-hop communication. a numerical example might make it clearer.

Response :The above correction was done in the revised manuscript.

  1. Figure 2 does not show a flowchart. Please explain the process/algorithm with concise reasoning (what condition is met before moving to social phase or individual phase or abandon phase?)

Response : Search and Rescue Optimization (SRO) for people typically take place in two distinct phases: social and individual.

Group members search for clues based on their location and quality in areas that are more likely to yield better clues in the social phase. There is no regard for where or how many clues have been found by others during the individual search phase. In general, clues fall into two categories:

  • Hold clue: one member of the search group is present and searches around it.
  • Abandoned clue: group members have found the clue and there is no one in that position. In other words, the human who found the clue has left it to find better clues, but the information about that clue is available for group members.

  1. There is no clear explanation of the methodology used for performance evaluation. 

Response : The performance validation was explained in the ection4.

  1. Line 286-288: The sentence is weekly related to the contents

Response : The above corrections were done in the revised manuscript

  1. Table 1 and Figure 3 have the same caption, similar to Table 2 and Figure 4.

                        Response : The above correction were done in the revised manuscript,

  1. What are FND, HND and LND. It appears that they are undefined.

First Node Dies (FND)

Half Node Dies (HND)

Last Node Dies (LND)

  1. Please thoroughly check the whole paper for errors such as (Line 137: dynamic node => dynamic nodes)

Response :  dynamic nodes Cleared

Round 2

Reviewer 1 Report

Some of my concerns have not been addressed. Such as,
1. There is a problem with the formula in line 242;  
2. The problem of subtracting different matrix dimensions in line 257 is not clearly explained;  
3. "X_j" in the formula at line 256 does not match "X_i" in line 258-259.  
4. The format of the formula in line 266 is wrong, and the meaning of "SEOR" is unknown;  
5. The format of lines 342-347 is wrong;  
6. The meaning of "MU" in line 316 is unknown.  

I am not satisfied with the figures in the paper, which need further improvement by the authors. 

Author Response

Response to Reviewers

Some of my concerns have not been addressed. Such as,
1. There is a problem with the formula in line 242;  

Response : Thanks for pointing this issue. We have mentioned DEG instead of NDEG. This issues has been corrected.
2. The problem of subtracting different matrix dimensions in line 257 is not clearly explained;  

Response: Thanks for pointing this issue. As per the reviewer comments the above equations(12) clearly explained with all the required parameters.
3. "X_j" in the formula at line 256 does not match "X_i" in line 258-259.  

Response : Thanks for pointing this issue. We have wrongly mentioned as Xi instead of Xj. Now this typo error was corrected.
4. The format of the formula in line 266 is wrong, and the meaning of "SEOR" is unknown;  

Response : Thanks for pointing this issue. In the formula   was typed SEOR. Now the error was corrected in line no 266 and 267 in the updated manuscript.
5. The format of lines 342-347 is wrong;  
Response: Thanks for pointing this. The above mentioned equation format was corrected in the updated manuscript.
6. The meaning of "MU" in line 316 is unknown.

Response : Thanks for pointing this , MU(Multi-user) was updated in the revised manuscript.

 All the diagram quality were improved with the required information in the standard format.

Thank you